# New Insights into the Relationship between Gut Microbiota and Radiotherapy for Cancer

**DOI:** 10.3390/nu15010048

**Published:** 2022-12-22

**Authors:** Zhipeng Li, Xiyang Ke, Dan Zuo, Zhicheng Wang, Fang Fang, Bo Li

**Affiliations:** 1NHC Key Laboratory of Radiobiology, School of Public Health, Jilin University, Changchun 130021, China; 2Key Laboratory of Carcinogenesis and Translational Research, Department of Radiation Oncology, Peking University Cancer Hospital and Institute, Ministry of Education, Beijing 100142, China; 3School of Public Health, Jilin University, Changchun 130021, China

**Keywords:** gut microbiota, cancer, radiotherapy, radiosensitivity, gut-organ axis

## Abstract

Cancer is the second most common cause of death among humans in the world, and the threat that it presents to human health is becoming more and more serious. The mechanisms of cancer development have not yet been fully elucidated, and new therapies are changing with each passing day. Evidence from the literature has validated the finding that the composition and modification of gut microbiota play an important role in the development of many different types of cancer. The results also demonstrate that there is a bidirectional interaction between the gut microbiota and radiotherapy treatments for cancer. In a nutshell, the modifications of the gut microbiota caused by radiotherapy have an effect on tumor radiosensitivity and, as a result, affect the efficacy of radiotherapy and show a certain radiation toxicity, which leads to numerous side effects. What is of new research significance is that the “gut-organ axis” formed by the gut microbiota may be one of the most interesting potential mechanisms, although the relevant research is still very limited. In this review, we combine new insights into the relationship between the gut microbiota, cancer, and radiotherapy. Based on our current comprehensive understanding of this relationship, we give an overview of the new cancer treatments based on the gut microbiota.

## 1. Introduction

Currently, the world is experiencing a huge cancer burden that is still growing [1]. Cancer is the main cause of death worldwide and the threat it poses to human health is frightening. Therefore, in-depth research on the relevant factors and mechanisms of cancer development has in no way stopped. The gut microbiome is home to a variety of commensal bacteria and other microorganisms living in the gut, where biofilm, a high-level structure in the gut, forms [2,3]. Recent advances in sequencing technology have elucidated that the gut microbiota has a complex role in cancer development and the body’s response to treatment. The close connection between the gut microbiota and cancer has become a new and exciting research hotspot. Often referred to as the “hidden organ”, the gut microbiota plays a crucial role in the body. It can affect a large number of physiological and pathological processes, including metabolism, vitamin synthesis, the integrity of the intestinal mucosal barrier, immune regulation, and protection against pathogens [4]. When various harmful factors act on the intestinal tract, causing disorder in the gut microbiota, they destroy the protective biofilm in the intestinal tract and may thereby affect the occurrence and progression of cancer in multiple systems. Experimental results have found that *Streptococcus* in early gastric cancer (GC) patients significantly increased and has a strong correlation with the occurrence of GC and liver metastases [5,6]. Ishaq et al. [7] conducted a study on the differences in gut microbiota between patients with esophageal cancer and unaffected people and showed that the intestinal bacterial composition of patients with esophageal cancer changed significantly. This phenomenon provides preliminary evidence to clarify the relationship between the gut microbiota and esophageal cancer. Surprisingly, the gut microbiota has also been found to be associated with extra-gastrointestinal tumors, such as breast cancer, leukemia, and lung cancer, although the relevant research is not sufficient to verify this link [8,9,10]. With more and more extensive and in-depth research, the relevant mechanisms of the gut microbiota affecting cancer are also gradually emerging. The gut microbiota might have an effect on the incidence and improvement of cancer through inflammation and immune regulation, signaling pathway activation, bacterial metabolite production, intestinal metabolism, and other aspects.

Radiotherapy is an effective method by which to treat tumors. The interaction between the gut microbiota and radiotherapy is a bidirectional function [11]. Radiotherapy will destroy the composition of the gut microbiota, which will have a positive or negative impact on the efficacy of tumor treatment. At the moment, there are still only a few studies on how the gut microbiota affects the efficacy of radiotherapy. In contrast, the relationship between the gut microbiota and the efficacy of immunotherapy and chemotherapy is clearer [12]. In this review, we summarize several mechanisms that have been identified that may play a role in the relationship between the gut microbiota and radiotherapy efficacy. In addition, changes in the gut microbiota, caused by exposure to ionizing radiation, can also lead to a series of side effects that affect the survival and quality of life of patients, such as radiation enteritis and diarrhea. What is more, the side effects are not confined to the gastrointestinal tract, but can also lead to sleep disturbances, cognitive impairment, and abnormal cardiopulmonary function, among other systemic problems. Recent evidence suggests that this may be related to the involvement of the gut microbiota in the “gut-organ axis”, which is formed by two-way communication between the gut and other organs [13]. Based on the above complex relationship between the gut microbiota and cancer, the gut microbiota has great potential to become the main choice for personalized cancer treatment strategies in the future, offering a new opportunity and challenge for cancer treatment [14]. In this review, we summarize new findings that target the gut microbiota for the treatment of cancer and the side effects of radiotherapy.

## 2. The Relationship between the Gut Microbiota and Cancer

There are many factors affecting the development of cancer. In addition to lifestyle, genetic and epigenetic changes, and environmental factors, it is gradually being proven that the composition and changes in the gut microbiota also make a difference [15]. The human gut microbiota is made up of 100 trillion archaeal and bacterial cells, distributed over more than 1000 species, and its impact on human health is clear. In healthy humans, the gut maintains homeostasis through interactions between the host and its microbiota, which prevents pathogens from invading [16]. However, changes in the gut microbiota can disrupt this balance and may lead to adverse host effects, such as tumorigenesis and progression [17]. Based on a large number of studies in the literature, we found that the gut microbiota not only plays a local role in the tumor development of gastrointestinal tissues but also has a distant effect on the tumor development of extra-gastrointestinal tissues. For example, metabolites in the gut microbiota, pathogen-related molecular patterns, and antigens derived from the gut microbiota can be transported to the liver via the hepatic portal vein and may have an impact on liver cancer. The gut microbiota itself can migrate to other closely related tissues and influence tumor progression [18]. Translocation of bacteria from the intestinal compartment to the pancreas demonstrated the gavage of mice with fluorescently labeled commensal bacteria [19]. In a word, the gut microbiota is related to tumorigenesis and the development of multiple human systems, such as the digestive system, genital system, blood system, and respiratory system (Table 1). Here, we summarize new perspectives on the relationship between the gut microbiota and cancer and the related mechanisms that have come to light in recent years.

To date, there have also been studies demonstrating a causal relationship between intestinal imbalance and cancer [25]. Therefore, more and more people have begun to further explore the mechanisms, although our understanding is still limited. Studies have proposed that gut microbiota biofilms have shown their potential as a major force in initiating and maintaining colorectal cancer (CRC) progression. A vivid bacterial driver–passenger model was introduced to explain the role of biofilms. In this model, bacterial drivers initiate CRC formation through genotoxicity, while bacterial passengers maintain the CRC process through metabolites [26]. Of late, it has been reported that biofilms also occur in urothelial carcinoma tissues, and it has been proven that biofilm formation is related to bladder cancer [27]. Several studies have reported that bacterial infection may affect the eradication of gastric adenocarcinoma, which may be related to the *H. pylori* biofilm [28].

By consulting and synthesizing the literature, here, we preliminarily summarize the finding that various factors can lead to the dysregulation of microbiota, thereby destroying the biofilms. This may also further affect the occurrence and development of cancer through inflammation and immune regulation, signaling pathway activation, bacterial metabolite production, intestinal metabolism, and other aspects (Figure 1).

Gut microbiota can promote the development of certain cancers, such as CRC, by stimulating chronic inflammation [29]. At present, the active role of the “inflammasome-microbiome axis” in the occurrence and progression of CRC has been confirmed [25]. Some researchers have proposed that a high-fat diet affects the progression of esophageal cancer by changing the gut microbiota, inducing inflammation, and expanding stem cells [30]. Both the enterotoxigenic *Bacteroides fragilis* and *Fusobacterium nucleatum* induced a tumorigenic inflammatory environment and increased the numbers of IL-17, IL-23, and neutrophils [31]. Different gut microbiota backgrounds lead to different gene expression profiles, which, in turn, affect immune cell typing and the prognoses of colorectal tumors [32]. However, it is important to note that while the vast majority of evidence suggests that inflammation caused by the microbiome has a carcinogenic effect, recently, individual microbes have been shown to be connected with anti-cancer inflammation [33]. At the same time, researchers have found that the activation of signaling pathways may also be a potential mechanism for the gut microbiota to affect the occurrence and development of cancer. For example, *S. enterica* and *Fusobacterium nucleatum* (*F. nucleatum*) were found to cause tumors by affecting the Wnt signaling pathway. Researchers also found that some *Helicobacter* spp. had pathogenicity that affected the TGF-β signaling pathway. In addition, *E. coli* and its related toxins were found to be naturally associated with the p53 and Wnt signaling pathways [29,34]. Other studies have demonstrated the role of *Helicobacter pylori* in accelerating the development of gastric cancer, while the Yes-associated protein 1, a key effector of the Hippo pathway, is also involved in this process [35].

In addition, the metabolism of the gut microbiota may also affect the progress of cancer (Table 2). To sum up, the gut microbiota may exert carcinogenic effects directly or indirectly, via its products, from three aspects. The first is that it causes DNA damage, changes gene expression, and increases proliferation. Another aspect is the promotion of protumorigenic microbial niches, including biofilms, while a third is to cause immune dysregulation. All of these aspects are at the core of carcinogenesis. Fusobacterial adhesins, including Fap2, RadD, and FadA, promote Fn aggregation, adhesion to dysplastic tissues, and biofilm formation through various mechanisms [31]. B2-colicin, produced by *Escherichia coli*, induced DNA damage and toxin-induced dsDNA breaks [36,37]. CagA, produced by *Helicobacter pylori,* is a cytotoxin that induces inflammatory pathways in gastric cancer and may also be involved in the tumorigenesis of CRC [38]. Recently, new discoveries have been made regarding gut microbiota metabolites. For example, gut microbial β-glucuronidase (βG) is found in certain gut microbial strains and has been shown to play a key role in promoting azoxymethane (AOM)-induced gut microbial dysbiosis and intestinal tumorigenesis [39,40]. New studies have shown that formate, as a gut-derived tumor metabolite associated with CRC progression, drives CRC tumor invasion by triggering the AhR signaling pathway, while increasing cancer stem cell potency and promoting CRC development [41]. Furthermore, intestinal microorganisms ferment insoluble dietary fiber to produce short-chain fatty acids (SCFAs), including butyrate, propionate, and acetate, which can act as energy substrates to link dietary patterns to the gut microbiota, thereby improving gut health [42]. As the only known SCFA with anticancer activity, butyrate plays an important role in human health [43]. Wang et al. [10] found that butyrate, produced by the gut microbiota, especially *Faecalibacterium*, was significantly reduced in the feces of patients with acute myeloid leukemia (AML), and that gavage of butyrate or *Faecalibacterium* delayed AML progression in mice. Studies have also found that the abundance of SCFAs and SCFA-producing bacteria was significantly reduced in CRC [42]. However, in addition to its anticancer effects, it has also been suggested that butyrate may promote carcinogenesis by increasing abnormal epithelial cell proliferation, suggesting that SCFAs may have both beneficial and harmful effects on human health [44,45]. Trimethylamine N-oxide (TMAO) is also a metabolite of the gut microbiota, the concentration of which has been found to be associated with a variety of health outcomes, including pancreatic cancer, primary liver cancer, and prostate cancer [46]. Recently, evidence has also been presented that microbial sulfur production is a potential cause of CRC, but the research is not extensive. Wolf et al. [47] showed that 142 bacterial genera all contain sulfur genes, and microbial sulfur metabolism genes are statistically associated with CRC. They also suggest that both organic and inorganic sulfur genes are associated with cancer and that organic sulfur metabolism genes may be the most important contributors of H_2_S in the human gut.

The host metabolism associated with the gut microbiota has also been proposed to participate in the development of cancer. Emerging evidence suggests that there is a very close relationship between the risk of colorectal cancer and the bile acid (BA)-gut microbiota (GM) axis [48]. It is worth noting that miRNA expression is regulated by the gut microbiota, which in turn affects the host transcriptome and thus the development of CRC [49]. Lu et al. [50] experimentally established that 27 hydroxycholesterol (27HC), cholesterol and other metabolites, as well as bacteria closely related to lipid metabolism, were significantly reduced in the thyroid cancer group. Among them, 27HC was significantly related to metabolism-related microorganisms (*g_Christenenellaceae_R-7_group*). In addition, functional pathways related to steroid biosynthesis and lipid digestion were inhibited in the same group.

## 3. Interactions between Gut Microbiota and Radiotherapy

The development of new molecular techniques, such as 16S ribosomal RNA (rRNA) sequencing, DNA sequencing, and metagenomics, and the use of germ-free mice have gradually revealed the greater role of gut microbiota in host homeostasis [2]. Different bacteria perform beneficial or harmful functions in the human body (Table 3) [51,52,53,54,55,56,57,58]. With the relationship between the gut microbiota and the progression of cancer becoming clearer, the important role of the gut microbiota in the current anti-cancer therapies, and its ability to affect the efficacy and toxicity of cancer treatment, has also been emphasized.

The gut microbiota is affected by a variety of factors. As one of these many factors, radiotherapy leads to dysregulation of the gut microbiota, which is often manifested as decreased abundance and diversity of gut microbiota, increased harmful microbiota (*Proteobacteria* and *Fusobacteria*), and decreased beneficial microbiota (*Faecalibacterium* and *Bifidobacterium*) [57,58]. Because of this property, the gut microbiota can be used as a novel biomarker of radiation exposure, which can complement traditional chromosomal aberration analysis and significantly improve biological dose assessment [59]. At present, many studies have examined the changes in the microbiota caused by radiotherapy in different cancers. El Alam et al. [60] found a significant change in the gut microbiome composition during pelvic chemotherapy and radiotherapy (CRT), with increases in *Proteobacteria* and decreases in *Clostridiales*, whereas after CRT, the gut microbiome composition changed, with increases in *Bacteroides* species. Additionally, Goudarzi et al. [61] evaluated that CRT significantly reduced species richness and increased the relative abundance of gut-associated taxa in oropharyngeal swabs of patients with HPV+ oropharyngeal squamous cell carcinoma (OPSCC) but had no effect on fecal samples.

### 3.1. Gut Microbiota and the Side Effects of Radiotherapy

Radiotherapy is the cornerstone of modern management methods of malignant tumors, but it can also cause damage to normal tissues and produce a variety of side effects, affecting the treatment results and the quality of life of patients [62]. Cancer and its treatment have a negative impact on the gut microbiota, often leading to ecological imbalance. This may reduce the patient’s response to treatment and increase the systemic toxicity of these therapies [63]. Radiation is defined as the stress source of the microbial ecosystem of the gastrointestinal tract. The radiation-induced imbalance of the gut microbiota has been hypothesized to be negatively correlated with mucositis, diarrhea, and fatigue caused by pelvic radiation in cancer patients [64,65]. However, recent studies have also identified the role of certain unique gut microbiota in resisting radiation side effects [66].

Pelvic radiotherapy often causes severe gastrointestinal adverse effects, which seriously hamper the long-term application of radiotherapy in tumor treatment [67]. Interestingly, accumulating evidence suggests a role for the human gut microbiota in the process of inducing gastrointestinal adverse effects by radiotherapy. At present, there are the following explanations for the mechanisms of radiation-induced intestinal injury: intestinal epithelial injury, intestinal microvascular changes, the immune mechanism, neuro-immune interaction, the gut microbiota, and many other factors [68]. The gut microbiota and its metabolites (especially short-chain fatty acids) play an important role in radiation-induced intestinal injury [58]. Some experts have shown that the gut microbiota plays a major role in regulating the systemic immune response (especially the inflammatory response), affecting gastrointestinal toxicity, as in the case of mucositis [12]. The gut microbiota regulates intestinal immunity, primarily by interacting with pattern recognition receptor (PRR) systems (mainly Toll-like receptors (TLRs)). In the gut, TLRs that are expressed by intestinal cells and dendritic cells recognize several MAMP molecules on the surface of bacterial cells, such as lipopolysaccharide (LPS) and peptidoglycan, and then regulate the immune response [69]. Until now, many studies have shown that the changes in short-chain fatty acids are also closely related to radiation-induced intestinal injury. Short-chain fatty acids are usually produced by the gut microbiota. The most common gut microbiota producing SCFAs mainly comprises anaerobic bacteria, including *Bacteroidetes*, *Bifidobacteria*, *Clostridium*, *Streptococcus,* and so on [70,71,72]. It is worth mentioning that systematic studies have found that microbial metabolites such as SCFAs (including butyric acid and propionic acid) can boost the extrathymic production of Treg cells, hinting that by-products may adjust the communication between intestinal microorganisms and the immune system and regulate the balance of the pro-inflammatory response [73]. Radiotherapy can change the bacteria that produce SCFAs, thereby causing the changes in SCFAs and affecting the appearance of many diseases, such as intestinal radiation injury [58]. However, the way in which SCFAs affect the occurrence of these diseases remains unclear.

Radiotherapy for abdominal, pelvic, and retroperitoneal malignant tumors can cause many adverse reactions, among which radiation enteritis (RE) is one of the most serious and common intestinal complications. The intestinal diseases involved are extensive, complex, and persistent, leading to difficult and ineffective treatment [74]. Nowadays, more and more studies show that an imbalance of the gut microbiota may contribute to the occurrence and development of radiation enteritis. Wang et al. [75] found that there was biodiversity disorder in RE patients with cervical cancer, which was characterized by means of drastically reduced α-diversity, accelerated β-diversity, a relatively greater abundance of *Proteobacteria* and *Gamma proteobacteria*, and a poorer abundance of *Bacteroides*. In patients with RE after radiotherapy, *Coprococcus* was significantly enriched prior to radiotherapy. In addition, studies have shown that high counts of *Clostridium IV*, *Roseburia*, and *Phascolarctoxin* are significantly associated with radiation enteritis. Steady-state intestinal mucosal cytokines related to regulating the gut microbiota and maintaining the intestinal wall are significantly reduced in radiation enteropathy [76].

With the increasingly clear relationship between the gut microbiota and radiation enteritis, more and more attention has been paid to the pathogenesis of radiation enteritis induced by radiation therapy in the context of the gut microbiota. The pathophysiological mechanism of gastrointestinal mucositis has been widely accepted as involving five stages [77]. Previous studies have additionally proven that the gut microbiota participates in the pathogenesis of radiation-induced gastrointestinal mucositis, which may be achieved by regulating the inflammatory process and oxidative stress, intestinal permeability, mucus layer composition, resistance to harmful stimuli, the epithelial repair mechanism, and the activation and launch of immune effector molecules [78]. To sum up, the mechanism of radiation enteritis can be described thus: radiation can cause tissue damage, which then leads to the up-regulation and amplification of inflammation. The gut microbiota has been found to participate in this process through two mechanisms: translocation and dysbiosis. Radiation destroys the intestinal barrier and mucus layer, leading to bacterial translocation and activating inflammatory reactions. The dysbiosis caused by radiation will affect local and systemic immune responses. In addition, ulceration and inflammation are exacerbated, due to the passage of microbial products through the ruptured epidermis [11]. Many studies have also speculated that a variety of metabolic processes of the host, including intestinal epithelial lipid metabolism and bile acid metabolism, both of which may be changed due to RE-related dysbiosis of the flora (Figure 2). It is widely recognized that the lipid bilayer plays an important role in maintaining the integrity of intestinal epithelium. The disruption of the intestinal epithelial barrier is the key factor affecting the progression of RE. Li et al. [79] established experimentally that the relationship between glycerophospholipids metabolism and RE activity was highly significant; the abundance change of *Dubosiella* and *Alistipes* performs a necessary position in the disorder of lipid metabolism in the process of disease; five pairs of bacterial lipid metabolites manifest the strongest functional correlation with RE. In conclusion, it can be inferred that the dysregulation of the gut microbiota and lipid metabolism may affect the progress of RE. Moreover, the link between microbial bile acid (BA) metabolism and the progression of inflammatory bowel disease (IBD) has recently been revealed by clinical metagenomics and metabolomics [80]. Bile acids in the human colon can be converted into deoxycholic acid and lithotomic acid under the action of intestinal microorganisms such as *Bacteroides intestinalis*, *Bacteroides fragilis*, and *E. coli*. When the balance of the microbiome is disturbed, the proportion of primary/secondary bile acids increases [81,82,83]. Fang et al. [84] suggested that the CPT-11-induced disturbance of bile acid metabolism may inhibit the production of IL-10, which in turn aggravates the high permeability of the mucosal barrier, while in the case of radiation enteritis, the relationship between gut microbiota, bile acid metabolism, and radiation enteritis needs further experimental research.

In addition to RE, diarrhea is also a common radiation-related intestinal injury. Some studies have observed that pelvic radiotherapy can cause changes in the community composition of the gut microbiota at the level of phyla, order, or genus [85]. Among them, the potential protective effects of *Bifidobacterium*, *Clostridium cluster XIVa*, and *Faecalibacterium prausnitzii* are reduced, and *Enterobacteriaceae* and *Bacteroidetes* are increased, which may especially lead to diarrhea [86]. There is also evidence that *Actinobacteria* and *Bacilli* levels are increased in patients with diarrhea after radiotherapy [87]. Some studies have also found that the alpha diversity of patients with diarrhea is significantly lower than that of patients without diarrhea, suggesting that diarrhea caused by CRT may be related to the imbalance of gut microbiota [64,75,87,88,89]. In addition, Atarashi and his colleagues reported that the oral administration of a combination of 17 *Clostridium* strains to adult mice can alleviate the allergic diarrhea of models [90], further strengthening the correlation between the gut microbiota and diarrhea caused by radiotherapy.

Nowadays, more and more studies have proposed that the microbial-mediated disturbance of intestinal mucosal homeostasis (dysregulation) is considered to be one of the pathogenetic factors of CRT-related fatigue. In recent years, some studies have found that compared with non-fatigue patients, the bacterial abundances of *Eubacterium*, *Streptococcus*, *Adlercreutzia*, and *Actinomyces* in fatigue patients after CRT increased significantly, and the abundances of the microbial sucrose degradation pathways also increased [91]. In addition, González-Mercado and others also found that patients with CRC who had concomitant symptoms reported more severe fatigue after CRT than those without symptoms. The abundance of *Bacteroidetes* and *Blautia2* differed between symptomatic and asymptomatic subjects [92]. At the same time, an increasing amount of new evidence shows that lifestyle intervention and the use of specific probiotics may be conducive to regulating the gut microbiota, and may mediate beneficial effects to improve fatigue, which also indirectly indicates the role of the gut microbiota in chemoradiotherapy-related fatigue [93]. It is worth mentioning that Xiao et al. [94] collected a multidimensional fatigue self-report scale and stool samples of head and neck cancer patients before and 1 month after radiotherapy. It was observed that there were significant differences in the gut microbiota patterns of patients with different degrees of fatigue across time. In the high-fatigue group, the relative abundance of the short-chain fatty acid-producing group is low, while the abundance of inflammation-related taxa is high. At present, some studies have preliminarily shown that the biological “gut-brain axis” mechanism may be related to sleep disorder symptoms. What is interesting is that the results of functional pathways highlight that inflammation may be likely to drive the gut-brain axis for fatigue related to cancer [94,95].

Brain metastasis (BM) is a common condition in 20–40% of cancer patients, which seriously affects the quality of life and survival rate of the patients. Nowadays, whole-brain radiotherapy (WBRT) has been widely used in the effective treatment of brain metastases. However, it cannot be ignored that a series of neuropsychiatric adverse reactions will occur in the process of WBRT, such as cognitive dysfunction (CD). Increasing evidence indicates that the pathogenesis of CD may be bound up with the gut microbiota and gut-microbiota-brain axis. Luo et al. [96] found that the composition of the gut microbiota between CD and non-CD phenotypes had changed, as established by 16S rRNA sequencing analysis. In addition, they also observed that the levels of bacteria Phylum*-Bacteroidete*, Class*-Bacteroidia* and Order*-Bacteroidales* decreased in the CD group, while the levels of Genus-*Allobaculum* increased after WBRT. Therefore, CD induced by WBRT may be highly correlated with an abnormal composition of the gut microbiota. Furthermore, it has been reported that Rs4702-A is related to the increased expression of FURIN and brain-derived neurotrophic factor (BDNF) in the serum and peripheral blood mononuclear cells (PBMC) of glioma patients after radiotherapy, both of which are closely related to cognitive function. This study also found that Rs4702-A was significantly related to an increase in enterotype type I and a decrease in enterotype type III in the stool samples of glioma patients after radiotherapy [97]. This also suggests that there may be a relationship between the gut microbiota and cognitive impairment caused by radiotherapy. However, what was obviously controversial is that Lensu and others found that although irradiation increased the abundance of the *Bacteroidaceae* family (*Bacteroides genus*) in the gut microbiota, it had no effect on gut microbiota diversity, systemic inflammation, and cognitive ability in mouse experiments. Therefore, the causal relationship between cognitive impairment and head irradiation may be much smaller than we have thought; instead, it is more likely to be due to the physiological and psychological effects of the disease itself [98]. Consequently, the functions of radiotherapy on the brain and whether the gut microbiota is involved still need more research to explore the topics in depth.

Except for gastrointestinal tract damage and the impact on the brain, radiotherapy can also cause damage and lesions in other organs; relevant studies show that this has a certain relationship with gut microbiota. Radiation pneumonia and fibrosis are very common in patients with chest tumors who are receiving radiotherapy, which can significantly reduce the survival rate and quality of life of patients. Chest irradiation can lead to the disorder of the lung and gut microbiota. It was found that phycocyanin intervention can regulate the pulmonary and intestinal microbiota to their normal state. This is characterized by a significantly decreased abundance of inflammation-related bacteria and the increased abundance of probiotics that produce short-chain fatty acids, which alleviates radiation-induced pneumonia and fibrosis [99]. Hence, we can speculate that radiation pneumonia and fibrosis may be related to the dysregulation of gut microbiota caused by radiotherapy. It is worth mentioning that at present, some research achievements have proven that the microbiota has an important function in terms of radiation-induced bone loss. Maier and others analyzed the relationship between fecal bacteria and bone mass, as well as body weight, through animal experiments. The results demonstrated that the thickness of the tibial cortex of irradiated mice with a traditional gut microbiota (CM) was reduced by 15%, while that of irradiated mice with a restricted gut microbiota (RM) was reduced by 9.2%. Correlation analysis determined the relationship between trabecular bone parameters and *Bacteroides massiliensis*, *Muribaculum* sp., and *Prevotella denticola*. The results showed that osteolytic injury caused by radiation was related to the type of bacterial indicator system composed of the gut microbiota [100].

### 3.2. Gut-Organ Axis

The complex interactions between the gut microbiota and the host immune metabolic system affect the functions of other organs, forming an “axis” between them. The “gut-organ axis” directly or indirectly regulates physiological metabolic processes, which has an important impact on health and disease. Nowadays, studies have widely counseled that the gut-organ axis may also be a potential mechanism for the occurrence and development of specific cancers and radiation damage, but relevant research is only in the initial stages. The following figure shows a brief description of several gut-organ axes and human diseases (Figure 3).

Oral and gut microbes are two major microbial ecosystems in the human body, both of which are physically and chemically linked [101]. The translocation and communication of flora between the oral cavity and the gut may result from the impairment of the gut-oral barrier. In general, barrier function is immature or impaired in neonates and elderly people [102]. The gut microbiota shifts toward the oral microbiome under conditions of low gastric acidity, further indicating the role of gut-oral barrier dysfunction on microbiota transfer [103]. In addition, enteric microorganisms can be transmitted via direct fecal-oral contact or indirect exposure via contaminated fluids and foods [104]. Studies have shown that the human hand microbiota has characteristics that are highly overlapping with the gut and oral microbiomes, suggesting that the human hand may be a vector for fecal-to-oral microbial transmission [105]. Furthermore, fecal-oral transmission is also commonly seen in immunocompromised individuals. In patients with head and neck cancer, poor oral hygiene conditions can further exacerbate the oral colonization of Gram-negative enterobacteria, which issue is closely associated with radiotherapy [106]. Overall, oral and gut microbes interact through gut-oral and fecal-oral pathways, which, in turn, can form and/or reshape the microbial ecosystems in both habitats, ultimately regulating the body’s physiological and pathological processes. The oral pathogen(s) can disturb the barrier function of the intestine and invade the gut mucosa, which causes intestinal dysbacteriosis and chronic inflammation, eventually resulting in inflammatory bowel disease (IBD) [107,108]. A number of studies have also found that a series of oral bacteria are closely associated with CRC, by analyzing the microbial composition of the oral cavity, colonic mucosal tissue, and fecal samples from patients with CRC [109], implying that the cross-talk between microbes may also be an important mechanism of tumorigenesis in the digestive system. Cui et al. [110] also demonstrated that the oral microbiome and intestinal microbiota synergistically influenced the efficacy and prognosis of radiotherapy for CRC.

Current studies demonstrate that microbiota-induced peripheral immune responses can be translated into neuroinflammation via the vagus nerve of the gastrointestinal tract and the vasculature throughout the body [111]. It is noteworthy that the gut and oral microbiota have been discovered to set off immune cells and propagate inflammation from the periphery to the cerebral parenchyma, thereby promoting the onset and development of neurodegeneration [112].

Recent emerging evidence has highlighted that the gut microbiota takes part in bidirectional communication between the gut and the brain [13]. Studies have proposed that the gut-brain axis is most likely an important factor involved in the initiation and progression of neurodegenerative diseases. Commensal and pathogenic enteric bacteria can produce lipopolysaccharides and amyloids, affecting the brain and immune system function. Gut microbiota dysbiosis can induce local and systemic inflammatory responses, mediated by immunity [113]. It has also been shown that changes in the gut microbiota caused by diabetes mellitus are strongly associated with alterations in the gut-brain axis activity and autonomic nervous system [114]. Besides the gut microbiota, recently, Calhoun and his team demonstrated that brain function is also linked to smoking-induced oral microbiota dysbiosis, possibly involving immunological and neurotransmitter signaling pathways [115]. Moreover, certain probiotics are widely used to improve central nervous system function, precisely due to their regulating effects on the gut-brain axis [116]. Up to now, the gut microbiota has been proposed to be associated with a variety of neuropsychiatric disorders, including autism, depression, anxiety, schizophrenia, and attention-deficit/hyperactivity disorder (ADHD). Boonchooduang et al. [117] reviewed the relationship between ADHD and the gut microbiota and suggested that pediatric and adolescent ADHD patients have potentially distinct gut microbial profiles. Dodiya et al. [118] found that a chronic stress-induced, gut-derived proinflammatory environment exacerbated the Parkinson’s disease phenotype through a dysfunctional gut-brain axis. Hemmings et al. [119] observed that the diagnosis of major depressive disorder was once additionally related to a multiplied relative abundance of *Bacteroidetes*. Currently, psychotropic medications also act on several taxa and then exert their therapeutic effects. Early life trauma may alter the composition of the microbiome, thereby promoting proinflammatory cascades and increasing the risk of developing post-traumatic stress disorder. These findings demonstrate that alterations in the gut microbiota may affect brain function and also confirm the critical role of the gut-brain axis in neurologically related disorders [120].

The available evidence suggests that there is also a close connection between the liver and the gut, with bidirectional communication, which is termed the “gut-liver axis”. Different mechanisms by which gut microbiota dysbiosis leads to liver injury have also been proposed over time. At present, the mechanisms of the relationship between gut microbiota and chronic hepatitis B (CHB) have been reported to include the following three aspects: (a) the increased permeability of the gut to microbiota and its metabolites, such as LPS; (b) normal bile acid metabolism and signaling are disrupted by gut microbiota; (c) colonization of the gut by invasive oral microbiota [121]. Further studies on the potential impact of the gut microbiota on CHB will provide new ideas and targets for its treatment. In addition, pathogen-associated molecular patterns can cross the intestinal barrier, leak into the liver via the portal vein, and, in turn, promote the progression of non-alcoholic steatohepatitis (NASH), which also exemplifies the role of the gut-liver axis [122]. Odontogenic infection with *Pseudomonas gingivalis* was found to promote the development of nonalcoholic fatty liver disease (NAFLD) and NASH in mice fed a high-fat diet, which may be associated with lipid accumulation, fibrosis, and inflammation in the liver [123,124]. It is also believed that chronic hepatitis C can alter the intestinal bile acid (BA) profile and lead to an imbalance of BA biosynthesis, eventually driving disease development, the mechanisms of which may involve the gut-liver axis [125].

### 3.3. Gut Microbiota and Radiotherapy Efficacy

There is growing evidence that the gut microbiota may affect the body’s resistance to anti-cancer treatments, including chemotherapy, immunotherapy, radiotherapy, and surgery [12,126,127,128]. There is a bidirectional interaction between the gut microbiota and these treatments, and different treatments will lead to different changes in the gut microbiota, which, conversely, will lead to distinct treatment responses [129]. Among them, radiotherapy, as an effective method to treat tumors, is also the same. According to current research data, the gut microbiota has a negative or positive influence on the effectiveness of tumor radiotherapy. Although the gut microbiota has been confirmed by many studies as a possible key to mitigating radiotherapy-related complications, the immediate impact of the gut microbiota on the effectiveness of radiotherapy remains to be profoundly understood, representing a meaningful and interesting area in radiology research [130].

Crawford and Gordon provided a new perspective on the relationship between the gut microbiota and radiosensitivity by reporting that the gut microbiota increased the radiosensitivity of lymphocytes and endothelial cells in the mesenchymal core of small intestinal villi in sterile mice that received whole-body irradiation [131]. A study by Cui et al. also showed that changes in the constitution of gut microbiota enhanced the susceptibility of mice to gamma-ray irradiation [132]. Another study found that whole-body irradiation enhanced the translocation of gut bacteria to mesenteric lymph nodes (LNs) in a mouse model of melanoma, leading to a stronger anticancer response [133]. There is preclinical evidence of the reduced efficacy of anticancer radiotherapy in patients exposed to broad-spectrum antibiotics. The alteration of the gut microbiota induced by antibiotics may be the key to this phenomenon [134]. Nevertheless, there is a negative association between the two in the limited data available. Yang et al. [135] found a high expression of FOXQ1 in CRC tissues and cells, and an increased expression of β-catenin and nuclear translocation, mediated by the up-regulation of SIRT1, which was beneficial to CRC-related intestinal pathological bacteria. This enhanced radio-resistance in CRC cells was positively correlated with adverse prognoses in CRC patients, suggesting an underlying link between the gut microbiota and radio-resistance. Modern pharmacological studies show that Shengmai turmeric powder can protect the gut microbiota from imbalances caused by radiotherapy, which partly explains the pharmacodynamic mechanism of Shengmai turmeric powder in radio sensitization, and indirectly reflects the negative correlation between intestinal microbiota imbalance and therapeutic effect [136].

Although there is growing testimony that the human gut microbiota works in radio resistance, in what way the gut microbiota impacts the response to radiotherapy still needs to be further studied and explored. Fast-induced adipokine (Fiaf) is normally secreted from the villous epithelium of the small intestine and is often inhibited by the gut microbiota, resulting in increased fatty acid storage in adipose tissues and the liver. Studies have shown that Fiaf deficiency leads to a loss of resistance to radiation-induced apoptosis in villous endothelial and lymphocyte populations [131,137]. It was found that grape polyphenols (GP) increased the expression of genes concerned with intestinal barrier function and Fiaf. In addition, GP significantly raised the growth rate of *Akkermansia muciniphila* and reduced the proportion of *Firmicutes* to *Bacteroidetes* [138]. In conclusion, we can speculate that alterations in the gut microbiota may influence the metabolism of Fiaf and, thus, affect radiation sensitivity. Enhanced autophagy may have considerable implications for the treatment of radiotherapy-resistant tumors. Some studies have found that X-ray irradiation can significantly induce autophagy in parental cells, while radiation-tolerant cells rarely undergo autophagic cell death, suggesting that cell radiation sensitivity is related to autophagic cell death [139]. Whether the gut microbiota is involved is also not certain. However, some researchers have found that *F. nucleatum* was plentiful in recurrent CRC tissues after chemotherapy, which promoted the resistance of CRC to chemotherapy. In terms of the mechanism, *F. nucleatum* activates the autophagy pathway and subsequently changes colorectal cancer chemotherapeutic responses by targeting TLR4 and MYD88 innate immune signaling and specific microRNAs [127]. Nowadays, there is increasing evidence for the role of vitamin D and the gut microbiota in radiotherapy responses. Radiotherapy caused the alteration of gut microbiota, ultimately affecting serum vitamin D levels and its distribution and metabolism in the body. In turn, the changes in vitamin D level affect radiotherapy resistance in patients, which may be related to the intestinal microenvironment, immune molecules in the intestine, intestinal microbial metabolites, and signaling pathways related to vitamin D receptors [140].

Radiotherapy, as a well-recognized treatment and palliative method for cancer, has also been believed to induce an antitumor immune response, so immunomodulation could be one way in which the gut microbiota affects the radiotherapy response [12]. The powerful immunomodulatory effects of radiotherapy (RT) include the cross-priming of tumor-associated antigen (TAA) and anti-tumor CD8+ T cells and the abscopal effect [141]. At present, our knowledge of the abscopal effect is still insufficient. In principle, ionizing radiation can provide tumor-specific antigens from dying cells, as well as the maturation stimuli essential for dendritic cells to activate tumor-specific T cells. Ionizing radiation can limit tumor growth outside the subject of irradiation, which is considered to be the abscopal effect [142]. However, whether the gut microbiota and its metabolites are involved in this effect, caused by radiotherapy, is not clear. As an antibiotic, vancomycin mainly acts on Gram-positive bacteria and is confined to the gut. Vancomycin has now been shown to enhance the antitumor immune response and tumor growth suppression induced by RT. This synergy relies on the cross-presentation of TTA to cytolytic CD8+ T cells and on IFN-γ. Therefore, it may be feasible to enhance the antitumor activity of RT by eliminating vancomycin-sensitive bacteria [141]. In addition, studies have shown that SCFAs produced by the gut microbiota not only act locally but can also be absorbed in the bloodstream to inhibit histone deacetylase activity, regulate gene expression in distal organs, and regulate the effector function of CD8+ T cells [63].

Radiotherapy not only affects cancer cells but also the tumor microenvironment (TME), composed of tumor blood vessels and immune system cells. It can cause the destruction of endothelial cells and can trigger an inflammatory response. Damaged vessels inhibit tumor infiltration by CD8+ T lymphocytes and activate the immunosuppressive pathways [143]. Interestingly, previous evidence has suggested that radiation simultaneously promotes a pro-immunogenic milieu within the tumor, which instigates cancer-specific immune responses in the host [144]. The delicate balance between the immune responses and immunosuppression activated by RT may play a critical role in efficacy [143]. Unexpectedly, the gut microbiota was identified as an important external regulator of TME [145]. It has been proven that it performs a vital function in tumor immunity by participating in the immune reprogramming of innate and/or adaptive immune cells to TME, which may have an effect on the efficacy of conventional chemotherapy and immunotherapy for pancreatic cancer [146,147,148]. However, whether this effect occurs in response to radiotherapy is unclear.

In vivo and clinically, the original immune effects generated by radiation are often much weaker than the immunosuppressive microenvironment characteristic of established cancers, so the original immune effects are overridden [149,150]. Nevertheless, when established immunosuppression is eliminated or diminished, such as in the addition of immune checkpoint inhibitors to local radiotherapy, the pro-immunogenic effects of ionizing radiation come into play and promote immune-mediated tumor resistance [142,151,152]. Immune checkpoint inhibitors (ICI) are an effective cancer treatment that rescues the activity of antitumor T cells in the tumors that are infiltrated by lymphocytes and are known as “hot” tumors. In contrast, “cold” tumors are refractory to immunotherapy because of the lack of lymphocytic infiltration. Preclinical data shows that radiotherapy can aggregate antitumor T cells and increase the sensitivity of refractory tumors to immune checkpoint inhibitors [153]. Recent studies on ICI therapies have shown that the changed gut microbiota may affect the antitumor immunity of tumor patients, thereby affecting the resistance to ICI, while the supplementation of different types of bacteria can restore the response to anticancer drugs [154,155]. It is worth mentioning that the gut microbiota can modulate antitumor immunity through small-molecule metabolites, which can disseminate from their original location in the gut to the whole body, influencing local and systemic antitumor immune responses and promoting the efficiency of ICI [146].

## 4. Radiotherapy for Cancer Based on the Gut Microbiota

### 4.1. Treatment of Cancer and Metastasis

The intestinal inflammatory and immunosuppressive microenvironment created by oral-gut microbes is a key factor in the development and progression of CRC [108]. Wang et al. [156] investigated the theory that sodium butyrate (NaB), a major product of gut microbial fermentation, could modulate the gut microbiota in mice with CRC liver metastasis (CLM) and improve the host immune response. These findings confirm that NaB is an effective treatment for CLM. Currently, Omega-3 polyunsaturated fatty acids (PUFAs) have been observed to significantly increase the density of bacteria known to produce butyrate, so they are thought to be useful to prevent dysbiosis of the gut microbiota and reduce the risk of developing CRC [157]. Corylin has been shown to reduce the risk of colitis-associated colon cancer (CAC) by modulating the composition of the gut microbiota, inflammation, and carcinogenesis, showing an effect of improving cancer. Therefore, as a natural agent, corylin may have potential functions against CAC [158]. Shang et al. [159] reported that chondroitin sulfate (SCS) in sturgeon might inhibit the growth of CRC by regulating the gut microbiota and altering the manufacture of certain amino acids. Liang et al. [160] demonstrated the development of a novel fasudil derivative, which can improve the gut microbiota and the function of the intestinal barrier in rats, and may act as a potent anti-breast cancer (BC) drug. Emerging studies have indicated that next-generation probiotics (NGPs) have beneficial effects in preventing carcinogenesis and provide new promising therapeutic options for cancer patients. Among them, the presence of *Faecalibacterium prausnitzii*, *Akkermansia muciniphila*, and *Bacteroides fragilis* in the digestive tract can affect the incidence of cancer [52]. Probiotics, prebiotics, and diets can also modulate the composition of the gut microbiota, which, in turn, protects patients with CRC from treatment-related adverse effects [161]. Meanwhile, probiotics are also widely recognized for preventing and treating breast cancer [162]. Yang et al. [163] suggested that a Quxie capsule (QX), a traditional Chinese medicine, could enhance CD4 T (TH) cell levels among metastatic colorectal cancer (mCRC) patients and increase the abundance of gut anticancer bacteria, such as *Actinobacteria*, as well as bacteria that produce butyrate, such as *Lachnospiraceae*. Many results indicated that the QX capsule may play a dual role in resisting tumors and enhancing immunity via microorganisms. Direct-contact moxibustion (DCM) used to be determined, in order to reduce *Ruminococcaceae* and *Prevotellaceae* bacteria (bacteria that produce SCFAs) and promote the growth of probiotic *Akkermansia*, thereby facilitating gastric mucosa apoptosis and delaying the development of gastric cancer [164]. This study also indirectly demonstrated the adverse effects of SCFAs on human health.

### 4.2. Prevention and Treatment of Radiation Injury

Mucositis and other gastrointestinal symptoms caused by radiotherapy resulted in a decreased appetite and insufficient nutrient intake. This is a non-negligible reason for the lack of nutritional diversity and the increased risk of malnutrition in patients. Multiple nutritional methods based on the regulation of gut microbiota may be a new way to alleviate radioactive gastrointestinal syndrome [165]. The therapeutic methods of gut microbiota modification in colorectal cancer include fecal microbiota transplantation (FMT), probiotics, and prebiotics. FMT plays a role in the following three aspects: the amelioration of bile acid metabolism, the modulation of immunotherapy efficiency, and the restoration of intestinal microbial diversity [166]. Probiotics are involved in the treatment of colorectal cancer by enhancing the host’s innate immunity, binding mutagens, and lowering the intestinal pH [154]. Prebiotics improve the intestinal microecology through the production of short-chain fatty acids and beneficial native stimulation [11]. Here, we briefly summarize the recent progress in the application of intestinal flora in treating various radiation injuries.

Oral mucositis (OM) is a common and unpreventable complication caused by chemoradiotherapy in patients with nasopharyngeal carcinoma. Chen et al. [167] evaluated that a combination of probiotics substantially enhanced the immune response in patients and acted on the composition of the gut microbiota, significantly decreasing the severity of OM in patients. The gut microbiota performs a function, primarily in the recovery segment of OM, by regulating the pro-inflammatory pathways [168]. Improving OM in the clinical setting by regulating the gut microbiota thus becomes more likely. Progress has also been made in the targeted treatment of the radiation-induced injury of other organs, based on the gut microbiota. Briefly, l-histidine and the imidazole propionate (ImP) produced by the gut microbiota have protective effects against radiation-induced lung and heart toxicity [169]. Fecal microbiota transplantation (FMT) can alleviate radiation pneumonia and improve lung function in mice [170]. Qin et al. [99] showed that phycocyanin could attenuate the lung inflammation and fibrosis induced by radiation via regulating the imbalance of the lung and gut microbiota caused by radiotherapy in mice. With regard to radiation damage to the digestive system, gut microbiota dysbiosis promotes inflammatory factor expression, impairs intestinal epithelial barrier function, and aggravates RE [171]. Urolithin A (UroA), a metabolite of the ellagitannin gut microbiota, can significantly reduce radiation-induced enterocyte apoptosis and improve the intestinal morphological structure and enterocyte regenerative capacity in irradiated mice [172]. Cai et al. [173] reported that the green tea polyphenol (-)-epigallocatechin-3-gallate (EGCG) could reverse the intestinal microbiota disorder caused by radiation, such as normalizing the ratio of *Firmicutes*/*Bacteroidetes* and increasing the abundance of beneficial bacteria. In addition, EGCG has been proven to alleviate radiation-induced intestinal injury (RIII). The findings have provided new insights into the alleviation of RIII by EGCG, revealing that EGCG can regulate the gut microbiota for the prevention and treatment of RIII. Gastrointestinal damage caused by radiation can also be alleviated by oral gavage of *A. muciniphila* [174]. A clinical study conducted by Majid et al. confirmed that partially hydrolyzed guar gum (PHGG) supplementation increased the *Bifidobacterium* counts and reduced the frequency of diarrhea after the end of pelvic radiotherapy [175]. Photobiomodulation (PBM), also known as low-level laser light therapy (LLLT) is a phototherapy that employs a low-level power light to specifically relieve pain and heal wounds [176]. Kiat et al. [177] elucidated that the PBM remedy brought about significant changes in the microbiome diversity of subjects, with an increase in the number of known beneficial bacteria and a decrease in the number of potentially pathogenic bacteria genera, suggesting that PBM may be a therapeutic strategy for chronic diseases. In conclusion, exploring the mechanisms of the gut microbiota in various tissues and organs after radiotherapy performs a necessary task in the prevention and treatment of radiation injury.

## 5. Conclusions and Future Direction

Based on all the above evidence, we have a primary perception of the complicated relationship between the gut microbiota, cancer, and radiotherapy (Figure 4). In view of this finding, we have launched many new cancer treatments targeting the gut microbiota, which will be a new opportunity and challenge for cancer treatments in the clinic. Phenomenally speaking, the critical importance of the gut microbiota in the incidence and improvement of various human cancers has been made clear. In terms of mechanism, gut microbiota disorder destroys biofilms, which may have an effect on the progression of cancer through inflammation and immune regulation, signal pathway activation, bacterial metabolite production, intestinal metabolism, and other aspects. It is worth noting that most of these mechanisms are found in CRC. However, the way in which the gut microbiota affects other cancers remains to be studied further.

The relationship between the gut microbiota and radiotherapy is complex and far more varied than the current studies have shown. Although more and more studies have proved the positive effect of the gut microbiota on the efficacy of radiotherapy, there is still some indirect evidence that a negative effect also exists. Similarly, while it is widely accepted that radiotherapy-induced alterations in the gut microbiota lead to severe side effects, the function of certain unique gut microbiota in protecting against radiation-induced side effects has recently been discovered. Consequently, further in-depth study of the relationship between the two is necessary. Based on the limited research, the way in which the gut microbiota affects the efficacy of radiotherapy and causes various serious side effects are not fully understood. Although some researchers have proposed that the gut microbiota may additionally play a function in RT-related immune responses, there is no direct evidence. Interestingly, several “gut-organ axis” may be associated with radiotherapy-induced side effects. This may be one of the most potential targets in the future on which we can rely to prevent the side effects of radiotherapy, while also broadening the exciting field of gut microbiota-based cancer therapy. 

## Figures and Tables

**Figure 1 nutrients-15-00048-f001:**
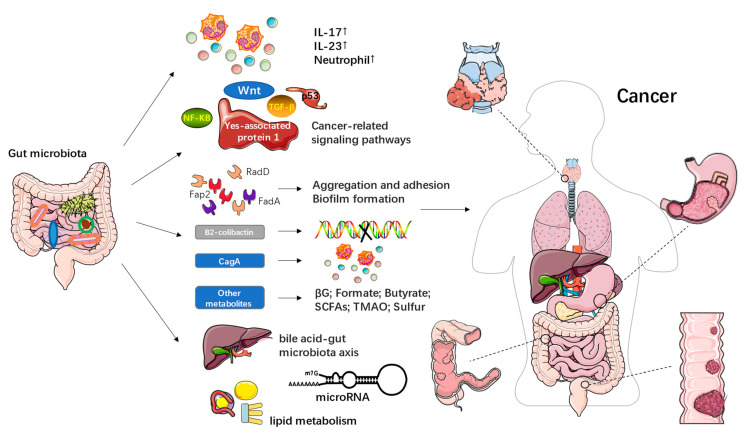
The possible mechanisms of the gut microbiota in the occurrence and development of cancer.

**Figure 2 nutrients-15-00048-f002:**
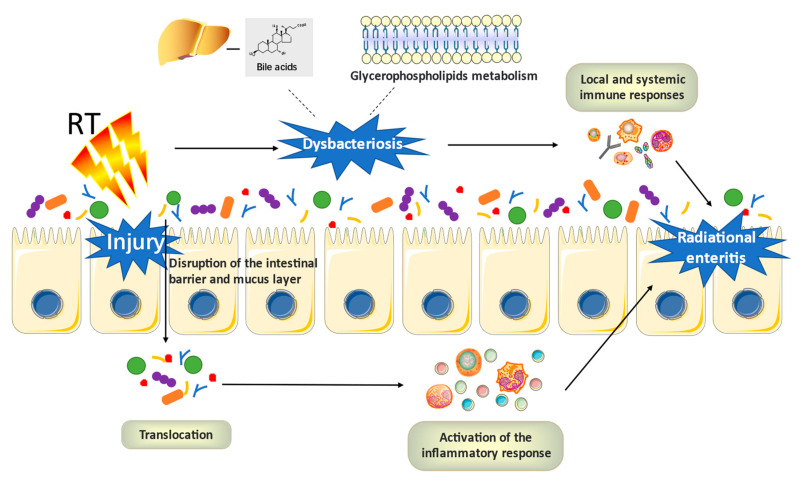
The potential mechanisms of radiation enteritis associated with gut microbiota.

**Figure 3 nutrients-15-00048-f003:**
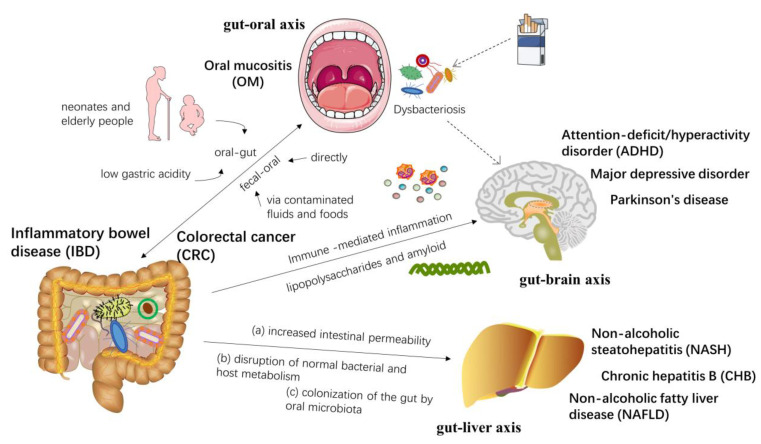
Several common “gut-organ axis” diseases and their interrelationships.

**Figure 4 nutrients-15-00048-f004:**
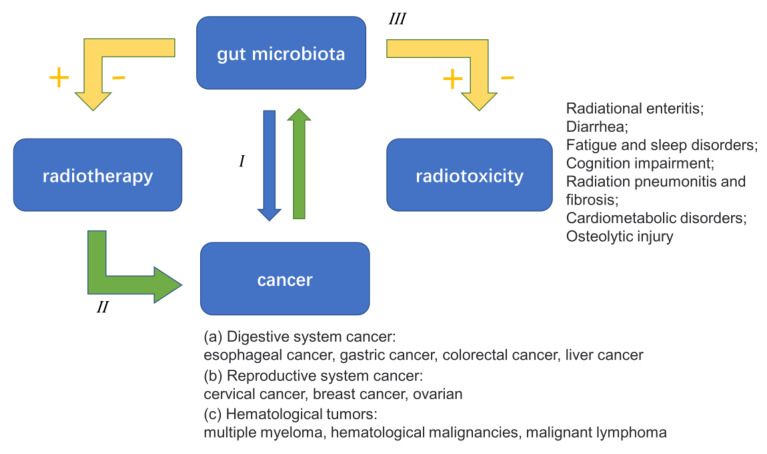
The complicated relationship between the gut microbiota, cancer, and radiotherapy.

**Table 1 nutrients-15-00048-t001:** Dysbiosis of gut microbiota in the development and progression of cancer.

Cancer	Object of Study	Sample Type	Method	Gut Microbiota Dysbacteriosis in Cancer Patients	Reference
colorectal cancer (CRC)	Brazilian CRC patients and a local control population	fecal samples	16S rRNA gene Amplicon sequencing	The α-diversity increased significantly; *Prevotella* increased and *Megamonas* and *Ruminococcus* decreased.	[20]
gastric cancer (GC)	45 GC cases from Shenyang, China.	cancer lesions and adjacent noncancerous tissues	ITS rDNA gene analysis	A significant increase of *C. Albicans* in GC; the abundance of *Fusicolla acetilerea*, *Arcopilus aureus*, and *Fusicolla aquaeductuum* were increased, while *Candida glabrata*, *Aspergillus montevidensis*, *Saitozyma podzolica*, and *Penicillium arenicola* were obviously decreased.	[21]
hepatocellular carcinoma (HCC)	40 healthy volunteers and 143 HCC patients	fecal samples	16S rRNA sequencing	Decreased α-diversity; a relatively lower average abundance of *Bacteroidetes* and a higher average abundance of *Actinobacteria* in the HCC group.	[22]
cervical cancer	42 cervical cancer patients and 46 healthy female controls	stool samples	16S rRNA gene sequencing	Higher alpha diversity in older women with cervical cancer; *Prevotella*, *Porphyromonas*, and *Dialister* were significantly enriched.	[23]
multiple myeloma (MM)	newly diagnosed patients with MM and healthy controls	fecal samples	deep metagenomic sequencing	Bacterial diversity was higher in MM; significantly enriched nitrogen-recycling bacteria in MM, such as *Klebsiella* and *Streptococcus*.	[24]
lung cancer (LC)	41 LC patients and 40 healthy volunteers	Stool and serum samples	16S rRNA gene sequencing and LC-MS analysis of serum samples	*Halanaerobiaeota*, *Actinomyces*, *Veillonella*, *Megasphaera*, *Enterococcus,* and *Clostridioides* were more abundant in the LC group.	[8]

**Table 2 nutrients-15-00048-t002:** Metabolites of gut microbiota and their effects.

Metabolites of Gut Microbiota	Effect in Cancer	Reference
Fusobacterial adhesins, including Fap2, RadD, and FadA	Promote Fn aggregation, adhesion to dysplastic tissues, and biofilm formation	[31]
B2-colicin produced by *Escherichia coli*	Induces DNA damage, and toxin-induced dsDNA breaks	[36,37]
CagA produced by *Helicobacter pylori*	Induces inflammatory pathways in gastric cancer; Involves in the tumorigenesis of CRC	[38]
Gut microbial β-glucuronidase (βG)	Promotes azoxymethane (AOM)-induced gut microbial dysbiosis and intestinal tumorigenesis	[39,40]
Formate	Drives CRC tumor invasion by triggering AhR signaling pathway; Increases cancer stem cell potency and promotes CRC development	[41]
Short-chain fatty acids (SCFAs), including butyrate, propionate and acetate	Link dietary patterns to gut microbiota; Butyrate has anticancer activity; Butyrate may promote carcinogenesis by increasing abnormal epithelial cells proliferation	[42,43,44,45]
Trimethylamine N-oxide (TMAO)	Associated with a variety of health outcomes, including pancreatic cancer, primary liver cancer, and prostate cancer	[46]
Microbial sulfur production	May be associated with colorectal cancer; Organic sulfur metabolism genes may be the most important contributors of H_2_S in the human gut	[47]

**Table 3 nutrients-15-00048-t003:** Harmful and beneficial bacteria commonly found in humans.

Beneficial Bacteria	Harmful Bacteria
*Acidophilus* *Akkermansia muciniphila* *Bifidobacterium* *Bacteroides fragilis* *Bacteroides thetaiotaomicron* *Christensenella minuta* *Clostridium casei * *Faecalibacterium prausnitzii* *Lactobacillus acidophilus* *Prevotella copri* *Parabacteroides goldsteinii* *Saccharomycetes* *Turicibacter* etc.	*Acetothermia* *Cryptomycota* *Desulfovibrio* *Deferribacteres* *Enterococcus* *Fusobacterium* *Lachnospiraceae* *Porhyromonas* *Pseudomonas* *Peptococcaceae* *Proteobacteria* *Rikenellaceae_RC9_gut_group* etc.

## Data Availability

Not applicable.

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
