# Peer review of "New Insights into the Relationship between Gut Microbiota and Radiotherapy for Cancer"

_nutrients, 2022, doi:10.3390/nu15010048_

Round 1

Reviewer 1 Report

Dear Authors, 

I evaluated your review of the New insights into the relationship between gut microbiota and radiotherapy for cancer. The pieces of information presented are well correlated and provide insightful aspects for better navigation on this tough topic. 

I suggest some minor revisions:

1. The table on page 3 needs to be reoriented, the entire text is not visible. 

2. The topic addressed is highly discussed in the last 2-3 years

·       https://doi.org/10.3389/fimmu.2021.622064; 

·       10.1007/s10120-021-01260-y; 

·       https://doi.org/10.1186/s13014-020-01735-9; 

·       https://doi.org/10.1155/2022/8183993; 

·  Cheng WYWu CYu J The role of gut microbiota in cancer treatment: friend or foe? 

As these works are highly similar to the topic presented in this manuscript (and are not cited in your review), an improved focus on originality is crucial. Please underline this aspect with priority.

3. The section Prevention and treatment of radiation injury should be extended with various nutritional approaches for gut equilibrium, stimulation with prebiotics, and immunomodulators, as well as some cautionary indications regarding the use of lactic fermented products during radiotherapy and the lack of diversity in nutrition for patients doing through this treatment.

Good luck! 

Reviewer 2 Report

Author suggested to make a table for bacterial metabolites and their effect with reference for easy understand.

Author suggested to list out the harmful and beneficial bacteria in human.

Author suggested to wright the conclusion very briefly.

Reviewer 3 Report

It has been a pleasure to review this manuscript. This review is well-structured and written. It compiles very up-to-date information and is well documented. Conclusions are well described and critical of the positive and negative effects of the gut microbiota on the efficacy of radioterapy and mitigate and prevent side effects derived from radiotherapy. Mainly refers to grammar corrections. These comments have been added to the attached file.
